# Is Repeat Abortion a Public Health Problem among Chinese Adolescents? A Cross-Sectional Survey in 30 Provinces

**DOI:** 10.3390/ijerph16050794

**Published:** 2019-03-05

**Authors:** Jinlin Liu, Shangchun Wu, Jialin Xu, Marleen Temmerman, Wei-Hong Zhang

**Affiliations:** 1Department of Public Health and Primary Care, International Centre for Reproductive Health (ICRH), Ghent University, 9000 Ghent, Belgium; liujinlin_xjtu@163.com (J.L.); Marleen.Temmerman@Ugent.be (M.T.); 2School of Public Policy and Administration, Xi’an Jiaotong University, Xi’an 710049, China; 3The National Research Institute for Family Planning (NRIFP), Beijing 100081, China; wu.shang.chun@yeah.net; 4Chengde Nursing Vocational College, Chengde 067000, China; xujialin1983@163.com; 5The Centre of Excellence in Women and Child Health, Aga Khan University, Nairobi 00100, Kenya; 6Research Laboratory for Human Reproduction, Faculty of Medicine, School of Public Health Université Libre de Bruxelles (ULB), 1070 Brussels, Belgium; 7Research Centre for Public Health, Tsinghua University, Beijing 100084, China; 8The INPAC Group, International Centre for Reproductive health (ICRH), Ghent University, 9000 Ghent, Belgium

**Keywords:** adolescent, repeat abortion, family planning, China

## Abstract

The Chinese Family Planning (FP) programme mainly focuses on married couples, and young unmarried women have limited access. This cross-sectional study aims to identify risk factors related to repeat abortions in Chinese adolescents receiving abortions. Data were collected using a questionnaire for all women seeking abortions within 12 weeks of pregnancy during a period of 2 months in 297 participating hospitals randomly selected across 30 provinces of China in 2013. Only the adolescents (younger than the minimum legal married age of 20 years) were included in this study. Of the 2370 adolescents who were receiving abortions, 927 (39%) were undergoing repeat abortions. The primary reason for the current unintended pregnancies was non-use of contraception (68%). Adolescents receiving abortions who had an increased risk of repeat abortions were those who had children (OR 2.57, 95% CI 1.80–3.67), those who resided in a middle-developed region (OR 1.81, 95% CI 1.30–2.50), those who resided in a relatively poor region (OR 2.40, 95% CI 1.78–3.23), and those who had used contraception during the 6 months preceding the survey (OR 1.38, 95% CI 1.12–1.71 for condom use). The occupation as a student was a protective factor for adolescents (OR 0.64, 95% CI 0.50–0.83). Adolescents should be offered equal access to FP to that of married women in China to reduce unintended pregnancies and repeat abortions. Correct and consistent contraception practice should be promoted.

## 1. Introduction

Unintended pregnancy is a global public health problem and is an extremely common occurrence in women’s lives [1]. For example, about half of all pregnancies in the United States were unintended [2]. Induced abortions are commonly used to end unintended pregnancies. A recent study estimated that 35 abortions occurred annually per 1000 women aged 15–44 years worldwide in 2010–2014 [3]. In addition, repeat abortion is another public health problem which cannot be overlooked and presents adverse effects on the reproductive and mental health of women, particularly for the young and unmarried women [4,5]. The identification of risk factors associated with repeat abortions and the development of public health initiatives to reduce the risk of repeat abortions recently witnessed growing international interest [6]. A recent systematic review by McCall et al. identified several determinants of repeat abortions, such as increased age, parity, marital status, contraception use at the time of conception and previous history of abuse or adverse life events [7].

In China, approximately six to ten million induced abortions have been reported annually since 2000 [8]. With dramatic social changes associated with rapid economic growth and reform over the past 30 years in China, traditional attitudes towards sex and sexual behavior have changed, and premarital sex has become more acceptable among young unmarried people, including adolescents [9,10,11]. Thus, unintended pregnancies and subsequently induced abortions among them increase. A previous study indicated that about 12% to 32% of unmarried women had become pregnant in China, among whom 86% to 96% had induced abortions [12]. A study conducted in 49 universities across seven cities in China reported that 31.8% of unmarried female students exposed to sexual activities in China had experience of unintended pregnancies, among whom 83.9% chose to terminate their pregnancies through induced abortions [13]. Liu et al. reported that 55.3% of 600 young unmarried women aged less than 24 years who were receiving abortions in Hebei were undergoing a repeat abortion [14]; and another study conducted on similar population in Beijing found that 26.9% of 1478 young unmarried women were undergoing a repeat abortion [15].

Numerous reasons result in the unintended pregnancies and induced abortions of young and unmarried women in China. The first reason may be the lack of sufficient programs regarding sex and contraception education in the traditional educational system of China, leading to their limited knowledge or even misconceptions about reproduction and contraception, especially when compared with their western counterparts [16,17]. To significantly reduce the incidence of unintended pregnancies among youth, the latest policy of a ‘Medium- and Long-Term Development Program (2016–2025) for young people (14–35 years)’ was proposed by the Chinese Central Government in 2017 to strengthen the popularization of sexual knowledge and implement sexual health education in conditional schools [18]. The second reason is linked to China’s family planning programme. Although it could provide related technical services (such as contraceptive education, pregnancy check-ups, and abortion services), it mainly focuses on the married couples [9,13], such that young unmarried people have limited access to those services such as contraception counseling and services including long-acting reversible contraceptive (LARC) services. Lack of counseling in family planning services prevents them from seeking contraception services [19]. Although condoms are readily bought and used, unintended pregnancies continue to occur due to their inconsistent or ineffective usage. A strong negative correlation between abortion and contraception use has been reported, such that abortion incidence declines as contraception use increase [20,21]. The provision of contraception by abortion providers, particularly the LARCs, can reduce the repeat abortions [6]. Prior studies in China have demonstrated that unintended pregnancies were directly related to non-use of contraception or ineffective contraception, and most of that had been terminated by induced abortions.

Adolescents are a critical target population with regard to global public health outcomes. However, they experience many sexual and reproductive health risks which are often attributable to early and unprotected sexual activities [22]. Compared with adults, they encounter more obstacles towards acquiring contraceptives, resulting in the low rates of their contraception use [23]. And a series of subsequent adverse outcomes are generated, such as unintended pregnancies and induced abortions. Adolescent pregnancies have more complications and are also unwanted in many cases [24]; and pregnancy-related morbidity and mortality are much more prevalent among adolescents than adults, particularly in low-income settings [25]. In 2009, the average abortion rate among all adolescents in the European Union reached 12.2 per 1000 girls aged 15 to 19 years [26]. Furthermore, similar to their adult counterparts, many adolescent girls might face risks of repeat abortions because of ineffective contraception following the prior abortions. Currently, very limited data exist on repeat abortions among adolescents globally. A nationwide retrospective register study in Finland reported that proportion of repeat abortions among adolescents seeking an abortion steadily increased from 8.8% in 1993–1995 to 14.3% in 2007–2009 [27]; the determinants for such increase included age, marital status, type of residence, and second-trimester abortion. In England and Wales, a study based on the data of national statistics on abortions found the percentage of repeat abortions among adolescents undergoing induced abortions increased from 9.1% in 1992 to 13.4% in 2013 [28]. In U.S., results of the Guttmacher’s 2000–2001 Abortion Patient Survey indicated that one in five adolescents obtaining abortions were undergoing repeat abortions [2]. In Kenya where safe abortion was restricted, one in every ten young women (12–24 years) seeking abortion-related care confirmed a previous induced abortion [29]; area of residence (urban versus rural), education, religion, employment, and contraceptive use at the time of current pregnancy were significantly influencing factors. In terms of Asian regions, data extracted from the Georgian Reproductive Health Survey 2010 showed that the percentage of repeat abortions among adolescents who had the experience of abortions was 20% [30], and another study conducted in three provinces in Vietnam found that 7.0% of adolescent abortion seekers were undergoing a repeat abortion [31].

In China, adolescents between 10–19 years of age accounted for almost 13.1% of the total population in 2010 [32]. A study in Shanghai verified a repeat abortion percentage of 38.5% among 2343 young unmarried women (≤24 years) undergoing induced abortions, with the determinants identified as age, occupation, education, the age of boyfriend, and cohabitation [33]. Another study conducted in three big Chinese cities reported that 33.0% of 4547 young unmarried abortion seekers (≤22 years) have had previous abortions, and education, migrant status, and contraceptive practice were significantly related to repeat abortions [34]. Nevertheless, to our knowledge, no nationwide study focused on repeat abortions among Chinese adolescents has been performed. In order to fill this literature gap, this study aims to describe the characteristics of Chinese adolescents undergoing abortions in China and to identify potential risk factors related to their repeat abortions.

## 2. Materials and Methods

### 2.1. Study Design and Participants

This cross-sectional study was a component of a collaborative research project for ‘Integrating Post-Abortion Family Planning Services in China (INPAC)’, which was funded by the European Commission (EC) under the Seventh Framework Program (FP7). Details of study design of the INPAC project have been published elsewhere [9,35]. The STROBE (Strengthening the reporting of observational studies in epidemiology) guidelines were followed when reporting results in this study [36].

The survey was carried out in 30 provinces (autonomous or municipalities) in Mainland China, with the exception of Tibet due to practical constraints on data collection [37]. A stratified cluster sampling design was used. A total of 300 medical institutions that provided abortion services were invited to participate in the survey according to the ‘criterion for medical institution level’ based on size (First-, second-, and third-level) and type (general hospital, maternal and children’s hospital (MCH), and family planning service institution) of institutions. Specifically, each province included ten medical institutions: two general hospitals and one MCH at the third level, two general hospitals and three MCHs at the second level, and one general hospital and one family planning service institution at the first level.

A total of 297 medical institutions were finally involved in this survey. Three were excluded because of unfinished data collection. During a period of two months, all the women seeking an abortion within 12 weeks of a pregnancy in the 297 medical institutions were invited to participate in the survey. In addition, only unintentionally pregnant adolescents who were younger than the legal minimum married age (20 years for women in China) and reported valid data regarding first versus repeat abortion were included in this research.

### 2.2. Data Collection and Variable Measurement

A structured questionnaire developed by the INPAC’s consortium was comprised of five sections: (1) General socio-demographic information, (2) Reproductive history, (3) Contraceptive use during the six months preceding survey, (4) Induced abortion history, and (5) Reason for current unintended pregnancy. The questionnaire was completed by abortion service providers in a paper-based or electronic format. Data were collected continuously for two months in each participating institution from March 20 to October 5, 2013. Finally, all data were incorporated and cleaned by the National Research Institute for Family Planning of China.

Six variables were used to measure the sociodemographic characteristics. (1) Age was initially a continuous variable and was divided into two categories: <18 and 18–19 years. (2) Education included seven categories: illiterate/semi-illiterate, primary school, junior middle school, senior middle school, senior college, university, and master and above. This study recoded a new variable with three categories: low-level (i.e., illiterate/semi-illiterate, primary school), medium-level (i.e., junior middle school, senior middle school), and high-level (i.e., senior college and above). (3) Occupation included 12 categories: jobless, housework, farmer, worker, business clerk, white-collar worker, civil servant/cadre, teacher/technician, domestic helper, private business owner, student, and other. This study recorded a new variable with five categories: jobless, housework, student, non-professional (i.e., farmer, worker, business clerk, domestic helper, private business owner, and other), and professional (including white-collar worker, civil servant/cadre, and teacher/technician). (4) Hukou status included five categories: local urban, local rural, non-native urban, non-native rural, and other (i.e., foreign and unclear). We recoded the variable of Hukou status into two distinct variables: residence status (i.e., rural, urban, and other) and migration status (i.e., migrant, non-migrant, and other). (5) The 30 provinces were classified into three categories of regions (i.e., relatively poor, middle-developed, and highly developed regions) according to the GDP (Gross Domestic Product) per capita of each province from 1997 to 2012 [38].

Reproductive history was measured by two variables: prior pregnancy and parity (number of children). The variable of contraceptive use during the six months preceding the survey included eight categories: non-use, rhythm, withdrawal, emergency contraception, condom, combined oral contraceptive pill (COC), and other. Induced abortion history was displayed by the number of prior induced abortions (including surgical abortion and medical abortion). For this study, we recoded it into a binary variable, i.e., repeat abortion, to measure whether the adolescents had an induced abortion prior to the current one (0 = no, 1 = yes). Reason for current unintended pregnancy included two categories: non-use of contraception and ineffective contraception, i.e., contraceptive failure.

### 2.3. Statistical Methods

Four continuous variables (i.e., age, number of children, number of prior pregnancies, and number of prior induced abortions) were tested for normality first. They all presented an abnormal distribution and were described using the “median” and “interquartile range (IQR)”. Categorical variables were described by “number” and “percentage.”

Pearson’s chi-squared tests were performed to assess differences in the proportions of socio-demographic characteristics, reproductive history, contraceptive use during the six months preceding the survey, and reason for current unintended pregnancy between adolescents undergoing repeat abortions and those undergoing the first abortions. We further estimated the crude odds ratio (OR) with 95% confidence interval (CI) for each variable except the self-reported reason of current unintended pregnancy. In addition, a multivariable logistic regression model was set up to explore the risk factors associated with repeat abortions of adolescents who were undergoing abortions. All variables in the model have been checked for interaction. The results were presented as the adjusted OR with 95% CI. All the significance levels were set at *p*-value < 0.05. The Statistical Package for the Social Sciences 24.0 (SPSS, IBM, Armonk, New York, NY, USA) for Mac was used for data analysis.

### 2.4. Ethics

Ethics approvals were obtained from both ethics committees of the National Research Institution for Family Planning (NRIFP), China, and of the Ghent University, Belgium (B670201421116). All participants signed a written informed consent of which they received a copy. The questionnaire was anonymous and the data were protected according to the European Commission regulations on Data Protection and Privacy guidelines.

## 3. Results

### 3.1. Participants

A total of 2370 adolescents were identified from 80,675 women who participated in the INPAC study. Figure 1 shows the profile of participants. Among the 2370 adolescents receiving abortions, the median number of prior induced abortions was 0 (IQR: 0–1; Range: 0–6). Specifically, 927 (39.1%) were undergoing a repeat abortion, and 206 (9%) for a third time or more. The proportion of adolescents receiving abortions without a prior induced abortion was 60.9%.

### 3.2. Univariable Analyses

The median age of these adolescents receiving abortions was 19 years (IQR: 18–19 years). 15.4% were less than 18 years, and the minimum age was 13 years. Among them, 78.9% received a medium-level education. Moreover, 31.7% were engaged in a non-professional occupation, 28.0% were jobless, and 25.5% were students. Over half (55.9%) were from rural areas, and 48.2% were migrants. In addition, 62.2% resided in relatively poor regions. A significant difference was observed between repeat and first abortion seekers with respect to education (*p* < 0.05), occupation (*p* < 0.001), and region (*p* < 0.001). In comparison to adolescents undergoing the first abortion, their counterparts receiving repeat abortions presented a significantly higher proportion in the following characteristics: low-level education, non-students, and residing in relatively poor regions (Table 1). Meanwhile, estimation of crude OR with 95% CI indicated that adolescents who had higher odds of receiving a repeat abortion were those who received a low-level education (OR: 1.99, 95% CI: 1.14–3.47, compared with “high-level education”), those who were engaged in housework (OR: 1.81, 95% CI: 1.18–2.79, compared with “jobless”), those who were non-migrants (OR: 1.19, 95% CI: 1.01–1.41, compared with “migrant”), and those who resided in less developed regions (OR: 2.47, 95% CI: 1.86–3.28 for relatively poor region; OR: 1.99, 95% CI: 1.46–2.72 middle-developed region, compared with “highly developed region”). However, adolescents who were students had higher odds of undergoing a repeat abortion (OR: 0.70, 95% CI: 0.56–0.89, compared with “jobless”).

The median number of children that adolescents had was 0 (IQR: 0–0; Range: 0–3). Almost all adolescents receiving abortions (93.0%) had no child, and only 7.0% had one child or more. The median number of prior pregnancies among adolescents was 0 (IQR: 0–1; Range: 0–6). More than half (56.3%) of adolescents receiving abortions did not have a prior pregnancy, 30.8% had one, and 12.9% had two or more. Significant differences were observed between adolescents undergoing a repeat abortion and those receiving a first abortion with respect to parity (*p* < 0.001) and the number of prior pregnancies (*p* < 0.001). Adolescents undergoing a repeat abortion presented a significantly higher proportion in the following groups: having one child or more and having one or more prior pregnancies (Table 2). Meanwhile, estimation of crude OR with 95% CI indicated that adolescents who had children (OR: 3.40, 95% CI: 2.43–4.75, compared with “having no child”) and had two or more prior pregnancies (OR: 4.55, 95% CI: 2.34–8.85, compared with “one prior pregnancy”) had higher odds of receiving a repeat abortion.

Results indicate that 62.0% of the adolescents receiving abortions did not use any contraceptive measures during the six months preceding the survey. Condom (58.9%) was the most common among adolescents receiving abortions who used contraceptive measures, followed by emergency contraceptive (13.4%), rhythm (10.8%) and other (16.9%, i.e., COC, withdrawal, and other). Among the repeat abortion seekers, 45.1% reported having used contraceptive measures, whereas those who were undergoing the first abortion and had used the measures only reached 33.4% (*p* < 0.001) (Table 2). Meanwhile, estimation of COR and 95% CI indicated that the adolescents who were receiving abortions and had used contraceptive measures during the six months preceding the survey had higher odds of undergoing a repeat abortion (OR: 2.47, 95% CI 1.63–3.75 for rhythm; OR: 1.52, 95% CI: 1.05–2.21 for emergency; OR: 1.39, 95% CI: 1.14–1.71 for condom; OR: 2.68, 95% CI: 1.75–4.12 for COC, compared with “non-use”).

The primary reason for current unintended pregnancies of adolescents receiving abortions was non-use of contraception (67.9%), followed by the ineffective contraception (32.1%). The current unintended pregnancies followed ineffective contraception for 37.9% of repeat abortion seekers and for 28.4% of the first abortion seekers (*p* < 0.001) (Table 2).

All results of estimation of COR and 95% CI for each variable above were displayed in Table 3.

### 3.3. Multivariable Analysis

In multivariable analysis, considering the high correlativity between the parity and number of prior pregnancies, and that no prior pregnancy would preclude a repeat abortion, we excluded the variable of number of prior pregnancies from the regression model (Table 3). Meanwhile, the variable of self-reported reason for current abortions was not included. After adjusting all remaining variables in the regression model, occupation, region, contraceptive use during the six months preceding the survey, and parity were significantly associated with receiving a repeat versus first abortion among adolescents undergoing abortions. Adolescents who resided in the less developed regions (OR: 1.76, 95% CI: 1.27–2.44 for middle-developed region; OR: 2.33, 95% CI: 1.72–3.15 for relatively poor region, compared with “highly developed region”), had used contraception during the six months preceding the survey (OR: 2.10, 95% CI: 1.32–3.33 for rhythm; OR: 1.56, 95% CI: 1.05–2.33 for emergency; OR: 1.30, 95% CI: 1.02–1.66 for condom; OR: 2.35, 95% CI: 1.47–3.75 for COC, compared with “non-use”), and had children (OR: 2.47, 95% CI: 1.70–3.58, compared with ‘no child’) had higher odds of receiving a repeat abortion. Meanwhile, adolescent abortion seekers who were students (OR: 0.67, 95% CI: 0.52–0.87) had higher odds of receiving a repeat abortion than those non-students. However, the variables of age, education, residence status, and migration status were not significant. In addition, we tested the bivariate interaction effects for all significant variables in univariable analyses, and no significant interaction effects were observed.

## 4. Discussion

To our knowledge, this was the first nationwide study that focused on the repeat abortions of Chinese adolescents receiving abortions. Meanwhile, the sample was very large and it was a part of a larger study which covered 30 provinces in China. The overall literature in this area at present was not enough to be done to distinguish. Almost all existing studies in China were conducted in only one or few cities, and most of the participants were women without any restrictions or the young unmarried women.

Our study found that 39.1% of the 2370 adolescents receiving abortions were undergoing a repeat abortion, which was similar to existing literature that reported repeat abortion percentage of young unmarried women in China ranging from 26.9% to 55.3% [14,15]. However, it was higher than that of adolescent abortion seekers in Finland, England and Wales, and the United States. 14.3% of adolescents seeking an abortion in Finland during 2007–2009 were receiving repeat abortions [27]; and the percentages of repeat abortions among similar population in the U.S. and England and wales were 13.4% and 20%, respectively [2,28]. No direct evidence was found to explain these differences. However, the gap of sexual education and contraceptive practice among adolescents between China and these developed countries may be relevant [16,17]. In response to that, the Chinese Central Government has proposed a programme called “Medium- and Long-Term Development Program for Young People (14–35 years)” in 2017 to strengthen sexual health education among young people and reduce young women’s risks of unintended pregnancies [18].

Results of our study indicated that 16 adolescents receiving abortions reported more than four previous abortions (with one respondent reporting as many as six), which was in line with the maximum number (4–8) reported in previous research conducted among young unmarried women in China [15,39,40]. In addition, the minimum age of adolescents undergoing a repeat abortion was 13 years in our study, which was lower than the finding of 15 years old in prior studies in China [40,41,42]. This outcome implies that very young women were already involved in sexual activities and were at risks of unintended pregnancies and subsequent induced abortions. If women have sex at younger ages, they are less likely to use contraception and at a greater risk of a pregnancy [34]. Therefore, this group of young age should attract more attention from governments and society.

In terms of the current unintended pregnancies of adolescents receiving abortions, 67.9% were related to non-use of contraception in our research, which was consistent with the results from prior studies. However, some studies reported ineffective contraception as the primary reason [15,34]. Additionally, more than three-fifths of adolescents receiving abortions in our study did not use any contraceptive measures during the six months preceding the survey. These results suggest extremely inadequate contraceptive practices among Chinese adolescents and further illustrate that adolescents were generally considered to exhibit risk-taking behavior with poor contraceptive practices [43].

Our study identified four factors that were significantly associated with repeat abortions among Chinese adolescents receiving abortions. The first factor was occupation. Results indicate that being a student was a protective factor. A decreased risk of repeat abortions was found among adolescents receiving abortions who were students compared with those who were jobless, which was consistent with prior studies in China [15,44]. Normally, adolescents in China aged 19 years or below should still be receiving an education in schools; however, only 25.5% of the adolescents in our study were still students. In comparison with their jobless counterparts with limited education, such adolescents possibly had more contraception knowledge and were more aware of the risks of unprotected sexual behavior.

The region was the second factor. One prior study reported that women with low socioeconomic status had an increased risk of repeat abortions compared with those of high socioeconomic strata [5]. Pradhan R. et al. reported that a higher risk of pregnancy among adolescents in Nepal was associated with living in the least resourced region [25]. In our study, although we did not directly measure the personal economic status of adolescents receiving abortions, we explored the effect of regions where they resided, which were classified according to the per capita of GDP and were representative of the general economic situation of adolescents. Findings indicate that residing in less developed regions (middle-developed region or relatively poor region) was a risk factor of repeat abortions of adolescents receiving abortions. Compared with the adolescent abortion seekers residing in highly developed regions, these adolescents who resided in middle-developed and relatively poor regions had 1.81 and 2.40 higher odds of undergoing a repeat versus first abortion, respectively. The less developed the region that the adolescents receiving abortions resided in, the higher the risk of undergoing a repeat abortion of them. This trend may be attributed to limited access to contraceptive measures and sexual health education and knowledge among these poor adolescents [30].

The third factor was contraceptive use during the six months preceding the survey. However, the importance of contraception with regard to repeat abortions of women is a controversial issue. In our study, compared with the adolescent abortion seekers who didn’t use contraception during the six months preceding the survey, these adolescents who had used contraceptive measures (rhythm, emergency, condom, and COC) had 1.38–2.58 higher odds of receiving a subsequent versus first abortion. Similarly, a study in Netherlands by Picavet et al. revealed that women who used combined hormonal contraceptive or long-acting method were 1.54 or 1.91 times more likely to have a repeat abortion than those who did not use contraception [45]. In another study conducted in the UK, Fisher et al. also found that women with repeat abortions were twice as likely to take oral contraceptives than women with first abortions [46]. However, the studies in the United States, Hungary, and Georgia documented no significance in terms of contraceptive use relative to repeat abortions [2,30,47]. The effective and correct use of contraception has real potential to decrease risks of unintended pregnancies and repeat abortions [48]. Excepting self-report bias, the results may indicate that although these adolescents were motivated to use contraception because of previous abortions, they were not consistent and correct users [15,34,45]. LARC use during adolescence is safe and most effective [26]; however, due to the restrictions of FP programmes in China, LARCs are mainly delivered through family planning clinics which only target married couples, but the majority of induced abortions are performed in public hospitals where Post-Abortion Family Planning (PAFP) services are often lacking and women who have undergone abortions are usually not referred to family planning clinics for FP counselling and services. The fragmentation of FP services is leaving a high risk to vulnerable groups such as young and unmarried women.

Parity was the fourth and a strong risk factor, which has been previously verified in the literature to have a positive effect on increasing the risk of repeat abortions in China and many other countries [2,5,30,45,47,48,49,50,51,52,53,54]. Women who had children were inclined to have repeat abortions compared with those without a child. In our study, 93.0% of the adolescents receiving abortions didn’t have a child; and compared with them, these adolescents who had one or more children had 2.57 higher odds of receiving a subsequent versus first abortion. Possibly, the latter was more casual on contraception and unintended pregnancy, and terminating unintended pregnancies through an induced abortion (even a repeat abortion) was more acceptable for them. Another reason might be that women with higher parity had repeat abortions because they did not want to look after another child [5].

Age, education, residence, and migration status were not significantly influencing factors related to repeat abortions of adolescents who were receiving abortions, which contradicted the results of prior studies on young women [15,27,30,34,42,45,47,48,49,50,51,55,56,57].

Our study has several potential limitations. First, as the sample in this study was adolescents who were receiving abortions, the results could not be generalized to all the adolescents in China. Second, as the survey is conducted by the way of self-reporting of the adolescents, this would bring a bias of social desirability, for example, adolescents undergoing a repeat abortion might especially feel like they should say they were using contraceptive methods (vs. no method use). Third, the study identified risk factors of repeat abortions among adolescents receiving abortions only using limited variables set in the questionnaire. Other factors that we did not collect and review (such as cohabiting, sexual abuse, age and contraceptive use at the first intercourse, characteristics of sexual partner, the education and occupation of parents, and personal income) are also important. Fourth, our data on contraceptive use were imprecise. Only a single choice was provided for participants when asking about which contraceptive measure they used before. This study overlooked the possibility of the joint use of multiple contraceptive measures. Furthermore, the kinds of contraceptive measure the participants used which led to the ineffective contraception (and resulted in their current unintended pregnancies and induced abortions) remain unidentified. At last, as the data were collected in 2013, there was a delay in dissemination of these results for 6 years. In future study, we aspire to conduct survey on samples of general adolescents and collect additional information to ascertain more risk factors of repeat abortions and determine the importance of contraceptive measures. Meanwhile, qualitative interview and research can be introduced to identify other possible predictors that are difficult to measure.

## 5. Conclusions

A large percentage of adolescents receiving abortions in China are receiving a repeat abortion, and the low self-reported contraceptive use of adolescents seeking abortions suggests a need for more contraceptive services for this age group. Furthermore, the abortion seekers of adolescents who were more likely to be undergoing repeat abortions were those who were jobless, those who had children, those who resided in middle-developed or relatively poor regions, and those who used contraceptive measures during the six months preceding the survey. Our study further emphasizes that promoting contraception and reducing unintended pregnancies among adolescents remain critical issues. Adolescents should be offered equal access to FP in China as that of married women to reduce unintended pregnancies and repeat abortions, for example, providing LARC services for adolescents. Correct and consistent contraception practice should be promoted.

## Figures and Tables

**Figure 1 ijerph-16-00794-f001:**
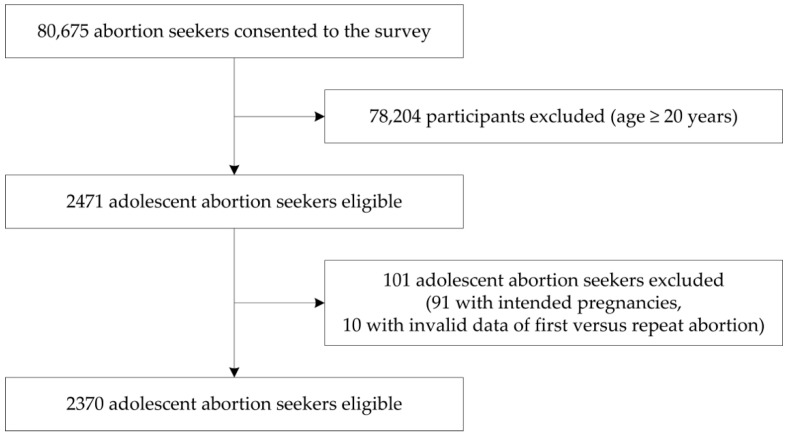
Study profile.

**Table 1 ijerph-16-00794-t001:** Sociodemographic characteristics of adolescents receiving abortions by history of abortion.

Characteristics	Total *N* (%)	First Abortion	Repeat Abortion	*p*-Value *
		*N* = 1443, *n* (%)	*N* = 927, *n* (%)	
Age (*n* = 2370)				0.162
<18 years	365 (15.4)	210 (14.6)	155 (16.7)	
18–19 years	2005 (84.6)	1233 (85.4)	772 (83.3)	
Education ^#^ (*n* = 2370)				0.047
Low	57 (2.4)	26 (1.8)	31 (3.3)	
Medium	1851 (78.1)	1128 (78.2)	723 (78.0)	
High	462 (19.5)	289 (20.0)	173 (18.7)	
Occupation (*n* = 2368)				0.000
Jobless	664 (28.0)	402 (27.9)	262 (28.3)	
Housework	96 (4.1)	44 (3.1)	52 (5.6)	
Student	605 (25.5)	415 (28.8)	190 (20.5)	
Non-professional	750 (31.7)	428 (29.7)	322 (34.7)	
Professional	253 (10.7)	152 (10.5)	101 (10.9)	
Residence status (*n* = 2367)				0.626
Rural	1323 (55.9)	816 (56.7)	507 (54.7)	
Urban	974 (41.1)	583 (40.5)	391 (42.2)	
Other	70 (3.0)	41 (2.8)	29 (3.1)	
Migration status (*n* = 2367)				0.119
Migrant	1140 (48.2)	718 (49.9)	422 (45.5)	
Non-migrant	1157 (48.9)	681 (47.3)	476 (51.3)	
Other	70 (3.0)	41 (2.8)	29 (3.1)	
Region (*n* = 2370)				0.000
Highly developed	308 (13.0)	784 (54.3)	417 (45.0)	
Middle-developed	588 (24.8)	359 (24.9)	231 (24.9)	
Relatively poor	1474 (62.2)	300 (20.8)	279 (30.1)	

* Chi-squared test; ^#^ Education: low-level (illiterate/semi-illiterate, primary school), medium-level (junior middle school, senior middle school), and high-level (senior college and above).

**Table 2 ijerph-16-00794-t002:** Reproductive history, contraceptive use in six months before survey, and reason for current unintended pregnancy of adolescents receiving abortions by history of abortion.

Characteristics	Total *N* (%)	First Abortion	Repeat Abortion	*p*-Value *
		*N* = 1443, *n* (%)	*N* = 927, *n* (%)	
Parity (*n* = 2370)				0.000
0	2205 (93.0)	1388 (96.2)	817 (88.1)	
≥1	165 (7.0)	55 (3.8)	110 (11.9)	
Number of prior pregnancies (*n* = 2370)				0.000
0	1335 (56.3)	1335 (92.5)	0 (0.0)	
1	731 (30.8)	98 (6.8)	633 (68.3)	
≥2	304 (12.9)	10 (0.7)	294 (31.7)	
Contraceptive use during six months preceding the survey (*n* = 2370)				0.000
Non-use	1470 (62.0)	961 (66.6)	509 (54.9)	
Rhythm	97 (4.1)	42 (2.9)	55 (5.9)	
Withdrawal	42 (1.8)	25 (1.7)	17 (1.8)	
Emergency	121 (5.1)	67 (4.6)	54 (5.8)	
Condom	528 (22.3)	304 (21.1)	224 (24.2)	
COC ^†^	92 (3.9)	38 (2.6)	54 (5.8)	
Other	20 (0.8)	6 (0.4)	14 (1.5)	
Reason for current unintended pregnancy (*n* = 2370)				0.000
Non-use of contraception	1609 (67.9)	1023 (71.6)	576 (62.1)	
Ineffective contraception	761 (32.1)	410 (28.4)	351 (37.9)	

* Chi-squared test. ^†^ COC: Combined oral contraceptive pill.

**Table 3 ijerph-16-00794-t003:** Binary logistic regressions on risk factors related to repeat abortions of adolescents receiving abortions.

Variables	Univariable Analyses		Multivariable Analysis ^c^ (*N* = 2365)	
	COR ^a^ (95% CI)	*p*-Value	AOR ^b^ (95% CI)	*p*-Value
Age				
<18 years	1.18 (0.94, 1.48)	0.154	1.20 (0.94, 1.52)	0.143
18–19 years	1		1	
Education ^d^				
Low	1.99 (1.14, 3.47)	0.015	1.43 (0.79, 2.62)	0.241
Medium	1.07 (0.87, 1.32)	0.524	1.00 (0.79, 1.27)	0.996
High	1		1	
Occupation				
Jobless	1		1	
Housework	1.81 (1.18, 2.79)	0.007	1.34 (0.84, 2.12)	0.221
Student	0.70 (0.56, 0.89)	0.003	0.64 (0.50, 0.83)	0.001
Non-professional	1.15 (0.93, 1.43)	0.185	1.10 (0.88, 1.37)	0.392
Professional	1.02 (0.76, 1.37)	0.898	0.94 (0.69, 1.28)	0.677
Residence status				
Rural	1		1	
Urban	1.08 (0.91, 1.28)	0.377	1.16 (0.96, 1.39)	0.131
Migration status				
Migrant	1		1	
Non-migrant	1.19 (1.01, 1.41)	0.043	1.00 (0.84, 1.21)	0.964
Other	1.20 (0.74, 1.97)	0.459	1.08 (0.64, 1.82)	0.767
Region				
Highly developed	1		1	
Middle-developed	1.99 (1.46, 2.72)	0.000	1.81 (1.30, 2.50)	0.000
Relatively poor	2.47 (1.86, 3.28)	0.000	2.40 (1.78, 3.23)	0.000
Contraceptive use				
Non-use	1		1	
Rhythm	2.47 (1.63, 3.75)	0.000	2.08 (1.35, 3.20)	0.001
Withdrawal	1.28 (0.69, 2.40)	0.434	1.29 (0.68, 2.45)	0.429
Emergency	1.52 (1.05, 2.21)	0.028	1.63 (1.10, 2.39)	0.013
Condom	1.39 (1.14, 1.71)	0.001	1.38 (1.12, 1.71)	0.003
COC ^e^	2.68 (1.75, 4.12)	0.000	2.58 (1.67, 4.00)	0.000
Other	4.41 (1.68, 11.53)	0.003	2.37 (0.86, 6.54)	0.094
Parity				
0	1		1	
≥1	3.40 (2.43, 4.75)	0.000	2.57 (1.80, 3.67)	0.000
Number of prior pregnancies				
1	1			
≥2	4.55 (2.34, 8.85)	0.000		

^a^ COR: crude odds ratio. ^b^ AOR: adjusted odds ratio. ^c^ Model fit information: *p*-value of Omnibus tests of model coefficients = 0.000, −2LL = 3007.702, Cox & Snell R^2^ = 0.065, Nagelkerke R^2^ = 0.088, *p*-value of Hosmer-Lemeshow goodness-of-fit test = 0.921. ^d^ Education: low-level (illiterate/semi-illiterate, primary school), medium-level (junior middle school, senior middle school), and high-level (senior college and above). ^e^ COC: Combined oral contraceptive pill.

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
