# Peer review of "Is Repeat Abortion a Public Health Problem among Chinese Adolescents? A Cross-Sectional Survey in 30 Provinces"

_ijerph, 2019, doi:10.3390/ijerph16050794_

Round 1

Reviewer 1 Report

Generally this is a well written paper covering an interesting topic. The age of the data (from 2013) is somewhat of concern, however. It is recommended that the authors utilize a guideline for reporting their results such as the STROBE guideline. Associations with repeat abortions are difficult to assess given that the study is cross sectional in nature, and the data collection not repeated, in addition to have been from more than 5 years ago, resulting in less siginificance of the findings for the reader. The term “floating” status is very odd, and should be changed to perhaps residency or migration status. Discussion section has some references to compare the results of this study to those of other countries, but tends to either compare with Europe/US or other studies from China, missing out those of other Asian regions. Would expect the Discussion to include more references to other Asian countries. Further the Discussion should include what specifically these results add to 1) the larger study which this was a part of and 2) the overall literature in this area—at present not enough is done to distinguish. The Limitations section needs to address the delay in dissemination of these results for 6 years.

Author Response

 Dear reviewer,

Many thanks for your review. Your comments and suggestions are very valuable to us, which contribute substantially to the refinement and improvement of our manuscript. We have studied them very carefully and try our best to revise the manuscript. A very detailed response to your each comment and suggestion has been made as below. Besides, we also make effort in English editing for improving our manuscript. Please find the revised versions with and without track changes in more details. The number of lines mentioned below corresponds to the revised manuscript without track changes.

Point 1: Generally, this is a well written paper covering an interesting topic. The age of the data (from 2013) is somewhat of concern, however. It is recommended that the authors utilize a guideline for reporting their results such as the STROBE guideline. Associations with repeat abortions are difficult to assess given that the study is cross sectional in nature, and the data collection not repeated, in addition to have been from more than 5 years ago, resulting in less significance of the findings for the reader. The term “floating” status is very odd, and should be changed to perhaps residency or migration status. Discussion section has some references to compare the results of this study to those of other countries, but tends to either compare with Europe/US or other studies from China, missing out those of other Asian regions. Would expect the Discussion to include more references to other Asian countries. Further the Discussion should include what specifically these results add to 1) the larger study which this was a part of and 2) the overall literature in this area—at present not enough is done to distinguish. The Limitations section needs to address the delay in dissemination of these results for 6 years.

Response 1: Many thanks. Following your suggestions, we 1) introduce the age of data and the delay in dissemination of the 6 years as a limitation (please see line 389-390), 2) organize the structure of our manuscript according to the STROBE checklist for cross-sectional studies and cite related literature of STROBE statement (please see line 126-127), 3) replace the “floating status” with the “migration status” across all the manuscript (please see line 164-165, Table 1 & Table 3), 4) add two studies from Georgian and Vietnam as it’s hard to find more studies which are directly related to the repeat abortions among adolescents in other Asian countries (please see line 106-110) , and 5) add that the study was a part of a larger study and overall literature in this area is not enough in the Discussion section (please see line 287-289).

Reviewer 2 Report

See attached file

Author Response

Dear reviewer,

Many thanks for your review. Your comments and suggestions are very valuable to us, which contribute substantially to the refinement and improvement of our manuscript. We have studied them very carefully and try our best to revise the manuscript. A very detailed response to your each comment and suggestion has been made as below. Besides, we also made effort in English editing for improving our manuscript. Please find the revised versions with and without track changes in more details. The number of lines mentioned below corresponds to the revised manuscript without track changes.

Part 1: Introduction

Point 1: Lines 56-58: The author(s) should be careful in citing percentages across the introduction, differentiating national data from any regional or less representative data. See cites 13-15

Response 1: Thanks. We add more details such as study setting and participants about these cited studies. Please see line 57-63 for details.

Point 2: Lines 60-67: Can you compare rates of sexual activity and contraceptive use (and which contraceptive methods are used) for teens in China compared to other countries – are there national data available for China?

Response 2: Thanks. Your suggestion is valuable and we also want to include these contents when we developed the paper. Unfortunately, there is no national available data and it’s hard to distinguish related data for this age group from the existing literatures, we are not able to make a reasonable comparison. We would introduce these contents in our future studies when it’s possible.

Point 3: Line 69: provide a cite and more detail here (and in the discussion) to better describe how China’s family planning program “mainly focuses on the married couples.”

Response 3: Thanks. We add two citations here and some other information about it. In china, the family planning program that focuses on married couples is related to its own policies’ regulations and restrictions. Please see line 73-76.

Point 4: Lines 75-76 notes that abortion providers provide LARCs in China. Can you assess this in this study? And, if not, it would be helpful to discuss potential family planning approaches to reduce repeat abortions in the discussion section (and how widely this is practiced after abortions).

Response 4: Thanks for your comments. This sentence is just to illustrate the relationship between repeat abortion and provision of contraception. As the restrictions of FP program in China, LARCs are mainly delivered through family planning clinics which only target married couples, but the majority of induced abortions are performed in public hospitals where Post-Abortion Family Planning (PAFP) services are often lacking and women who have undergone abortion are usually not referred to family planning clinics for FP counselling and services. The fragmentation of FP services is leaving a high risk to vulnerable groups such as young and unmarried women. Please see line 75-76, 357-363, 403-404.

Point 5: Lines 86-95: This section could be streamlined and clarified. My sense is that your main finding is that repeat abortions are less common outside of China, but that isn’t clear. Also, it would help to show percentages and populations comparable to the current study. The text mixes rates (which I assume are to all teens in line 87 and not to those undergoing abortions) with percentages of repeat abortions (line 91 - Finnish adolescent data shows percentages – is this among all teens or among women seeking an abortion?) The percentages for England and Wales in lines 93-94 reports percentages among women undergoing abortions (similar to this study). The U.S. data in lines 94-95 suggests longitudinal data among women who received abortions – is that correct?

Response 5: Thanks. This paragraph is meant to introduce the existing studies on abortions especially the repeat abortions among adolescents; however, such kind of studies are very limited. Here we don’t make comparisons but do them in the discussion section. As our study focuses on repeat abortions of adolescents, firstly, we just use EU’s data to introduce the abortion rate among adolescents, and you are right this result is based on all teens and we add this (please see line 92-93). Then, we present the situation of repeat abortions among adolescents based on existing studies. Sorry for the confusion brought by our previous introduction. The samples in the Finland’s study, England and Wales’ study, and US’ study are adolescents receiving an abortion, we rewrite these more clearly. Please see line 96-103.

Point 6: Line 115: Citation 33 about the program study was not accessible (listed as “temporarily disabled”), so I wasn’t able to access information about the study design – please check the link.

Response 6: Thanks. It’s our negligence. As the INPAC project has been completed and we don’t have additional funds to cover the costs of running and keeping the website, so it’s no longer available since last year. However, based on the project, we have made many academic achievements, such as book and papers. Here we add two citations that one is a book in Chinese  with English abstract and another one is an abstract published in the Lancet. Please see line 125-126.

Part 2: Measures

Point 7: Line 140: Since only 2.4% of the sample <=15, I would strongly suggest using a different cut- point. For U.S. studies, we often divide this into <18 and 18-19.

Response 7: Thanks. Following up to your suggestion, we change the cut-point from 15 years to 18 years. All related analyses have been revised. Please see line 151-152, 211-212, Table 1 & 3.

Point 8: Did you include measures of relationship status? Were any participants cohabiting? Married (although it sounds like they need to be 20 to have a legal marriage)? This seems particularly relevant given your introduction’s focus on nonmarital pregnancies.

Response 8: Thanks. All adolescents included in our study are unmarried. And as we don’t consider other relationship status such as cohabiting during our survey. And in the introduction and discussion sections, we have tried our best to focus on unmarried samples, especially for these studies from China. We also have mentioned this relationship status as a limitation. Please see line 382.

Part 3: Figure 1, data collection

Point 9: My main concern is that Figure 1 suggests that every eligible adolescent consented to the study and completed a survey. Is that correct? Or did the study approach any adolescents who did not consent to the study? If so, the sample may not reflect the sampled population (e.g., if certain groups did not agree to complete a survey). Also, did the study provide incentives?

Response 9: Thanks. As our study is a part of a larger survey which includes all women seeking an abortion during the study period, you can see from Figure 1, firstly 80675 abortion seekers consent to our survey and complete it, then we extract related data about eligible adolescents for including in this study. The abortion seekers who don’t consent to our survey are not invited to complete the questionnaire. However, during the investigation, almost all abortion seekers consent to our survey and complete the questionnaires. We don’t provide any incentives for them. We have revised the Figure 1 and please see line 203.

Part 4: Tables

Point 10: Education level (in Tables) should better describe each category. E.g., Low (illiterate/semi- illiterate, primary school). You can either provide this in the label or in a footnote.

Response 10: Thanks. We add the description of education level in footnotes after the tables. Please see Table 1 & 3.

Point 11: Table 3:

(1) The title should clearly describe the sample: Binary logistic regressions on risk factors related to repeat abortions of adolescents receiving abortions

(2) I don’t think you need Model 3

(3) Please add your final sample size to Model 2 – did you lose any of your sample to missing data? Also add model fit information.

(4) What did you do with “other” (for floating status)?

(5) Check for collinearity of contraceptive method and reason for unintended pregnancy – both have non-use as the reference category. I would use self-reported reason for abortion as a descriptive measure and not include it in your multivariate analyses.

Response 11: Thanks. (1) We revise the title of Table 3 according to your suggestion, the titles of Table 1&2 have also been revised clearly; (2) following your suggestion, we remove the Model II in Table 3; (3) the final sample size is 2365 in the multivariable analysis, and we add the N in Table 3; the model fit information has been provided in the footnotes after Table 3; (4) we add the results of “other” related to migration status in Table 3; (5) we check the collinearity of contraceptive use and reason for unintended pregnancy. The results show that the value of tolerance and VIF is 0.719 and 1.391, respectively, which indicate that there is no significant collinearity between them. However, your suggestion is very useful, so we remove the variable of self-reported reason for abortion from the multivariable regression model and just analyse it as a descriptive measure. Please see Table 3 and related description for more details.

Part 5: Results

Point 12: Throughout the results section (and discussion), it’s important to note that the sample is women receiving abortions. E.g., ** factors were associated with receiving a repeat vs. first abortion or with receiving a repeat abortion among women receiving abortions. Otherwise, a reader could confuse your findings as though you have longitudinal data on women who received an abortion.

Response 12: Thanks. We have revised all related description of the sample as the adolescents receiving abortions in our study. Sorry for the confusion brought by our previous description.

Point 13: Generally, logistic regressions are described as “higher odds” or a repeat abortion vs. “more likely to have”

Response 13: Thanks. Following your suggestion, we have revised all related description about the results of logistic regressions in our paper.

Part 6: Discussion

Point 14: Can you compare the % of abortions that were repeat among older women to those of teens in this study? In other studies?

Response 14: Thanks for your suggestion. The comparison has been considered in other studies in the future.

Point 15: Line 276 and elsewhere: clarify your population to: of the 2370 adolescents receiving an abortion. Line 279 About 20% of adolescents who were undergoing/receiving an abortion. Otherwise, it sounds like you have longitudinal vs. cross-sectional data

Response 15: Thanks. We have revised all the description of the samples in our study and related literatures more clearly.

Point 16: Line 284: similar to above, provide information on the % of teens in China vs. U.S. or other countries using effective contraceptive methods (hormonal or LARCs).

Response 16: Thanks. As no clear data and existing national study in China related to this, we could not provide the information. In the future, we will try to advance these studies in China.

Point 17: Lines 292-294: this statement overstates the study findings – having some 13-year-olds in the study does not necessarily represent a trend. Also, it’s not clear what you mean in lines 295-296 - could rewrite to say that if women have sex at younger ages, they are less likely to use contraception and at a greater risk of a pregnancy. This paragraph should present information on trends in sexual activity among teens in China – can you cite national data?

Response 17: Thanks. You are right that the statement overstates the findings. Following in your suggestions, we rewrite this paragraph to make it clearer and more reasonable. However, as there is no national data about the trends in sexual activity among teens in China, we are not able to present this information here. Please see line 309-312.

Point 18: Lines 307-08: I’m not clear what this means. Are you making a recommendation that adolescents in China should be in school? Or are you just saying that most teens in China are in school?

Response 18: Thanks. Sorry for the confusion. Here we just want to say adolescents aged 19 years or below normally should still be receiving an education in schools and introduce what we find in this study, not to make a recommendation. Please see line 324-326.

Point 19: Lines 320-322: Did you test whether there was a difference between middle-developed and poor-developed regions in Table 3? This sentence suggests that is the case.

Response 19: Thanks. Yes, we have tested the effect of region in the multivariable analysis. Results have been presented in Table 3. From the results, we can see the difference between middle-developed and relatively poor regions compared to highly developed regions. We add more details about the results of region in this paragraph, please see line 337-339.

Point 20: Line 326: this sentence suggests that you have longitudinal data. I would update to “had 1.34-2.45 higher odds of receiving a subsequent versus first abortion” or else add “among women receiving an abortion”

Response 20: Thanks. We have revised this following your suggestion, please see line 347.

Point 21: Lines 349-361: Limitations should include a statement about self-reports and social desirability. For example, women receiving a repeat abortion may especially feel like they should say they were using contraceptive methods (vs. no method use).

Response 21: Thanks. Following your suggestion, we include this in the limitation section. Please see line 378-380.

Part 7: Conclusion

Point 22: Line 363: It’s not clear that repeat abortions are “highly prevalent” – I would rework to say “a large percentage of women receiving abortions in China are receiving a repeat pregnancy.” Also, the statement “the contraceptive practices of that age group were inadequate” is not a conclusion of this study. You would need to describe contraceptive use among teens more generally to make this statement. Instead, you could say something like: “the low self-reported contraceptive use of women seeking abortions suggests a need for more contraceptive services for this age group”

Response 22: Thanks. We have rewritten this part according to your suggestion. Please see line 396-398.

Round 2

Reviewer 1 Report

I am ok with the revisions that the authors have made. I do not need to fully review again, as long as editor verifies the changes at the specific line numbers were made in track changes on the revision.